# Menadione Potentiates Auranofin-Induced Glioblastoma Cell Death

**DOI:** 10.3390/ijms232415712

**Published:** 2022-12-11

**Authors:** Monika Szeliga, Radosław Rola

**Affiliations:** 1Department of Neurotoxicology, Mossakowski Medical Research Institute, Polish Academy of Sciences, 5 Pawińskiego Str., 02-106 Warsaw, Poland; 2Department of Neurosurgery and Paediatric Neurosurgery, Medical University of Lublin, 8 Jaczewski-ego Str., 20-090 Lublin, Poland

**Keywords:** glioblastoma, thioredoxin reductase 1, auranofin, menadione, reactive oxygen species, drug repurposing

## Abstract

Glioblastoma (GBM) is the most aggressive primary brain tumor. Recently, agents increasing the level of oxidative stress have been proposed as anticancer drugs. However, their efficacy may be lowered by the cytoprotective activity of antioxidant enzymes, often upregulated in neoplastic cells. Here, we assessed the mRNA and protein expression of thioredoxin reductase 1 (TrxR1), a master regulator of cellular redox homeostasis, in GBM and non-tumor brain tissues. Next, we examined the influence of an inhibitor of TrxR1, auranofin (AF), alone or in combination with a prooxidant menadione (MEN), on growth of GBM cell lines, patient-derived GBM cells and normal human astrocytes. We detected considerable amount of TrxR1 in the majority of GBM tissues. Treatment with AF decreased viability of GBM cells and their potential to form colonies and neurospheres. Moreover, it increased the intracellular level of reactive oxygen species (ROS). Pre-treatment with ROS scavenger prevented the AF-induced cell death, pointing to the important role of ROS in the reduction of cell viability. The cytotoxic effect of AF was potentiated by treatment with MEN. In conclusion, our results identify TrxR1 as an attractive drug target and highlights AF as an off-patent drug candidate in GBM therapy.

## 1. Introduction

Glioblastoma (GBM), World Health Organization (WHO) grade IV, is the most frequently occurring malignant primary brain tumor. The median survival time of GBM patients is estimated to be less than 12 months, and the five-year survival rate is 6.8% [1]. GBM is characterized by uncontrolled cellular proliferation, necrosis, brain infiltration, and neoangiogenesis. Moreover, significant intratumoral heterogeneity at the genomic and cellular level caused by genomic instability and the presence of GBM stem cells (GSC) contributes lack of effective therapeutic options in GBM [2].

Drug repurposing, also known as drug repositioning, is the process of exploring new medical uses for existing drugs that are approved in clinical practice [3]. Recently, a number of licensed non-cancer drugs with evidence of anticancer activity have been included in a database (ReDO_DB) developed by the Repurposing Drugs in Oncology (Re-DO) project [4]. Auranofin (AF), approved for the treatment of rheumatoid arthritis, has received increasing attention as a potential anticancer drug [5]. AF is an orally administrated, gold(I) containing compound considered safe for human use [5]. The main mechanism of action of AF is related to its inhibitory activity against thioredoxin reductase (TrxR) [6,7]. Indeed, comprehensive chemical proteomics confirmed TrxR as a main AF target [8]. TrxR, together with thioredoxin (Trx) and NADPH, forms the thioredoxin system, which plays an essential role in maintaining the redox balance. In the brain, the Trx1 system, coupled with peroxiredoxin 1 (PRDX1) and 2 (PRDX2) and methionine sulfoxide reductase (MSR), regulates redox signaling in the cytosol and nucleus. The Trx2 system, coupled with peroxiredoxin 3 (PRDX3) and 5 (PRDX 5) and MSR, regulates the mitochondrial redox environment and signaling [9].

Increased oxidative stress caused by a high metabolic rate and accelerated prolifera-tion leads to a redox imbalance, which, in turn, contributes to GBM progression [10] and resistance to treatment [11]. To maintain the redox homeostasis, GBM cells overexpress antioxidant systems and regulators to counteract the increased oxidative stress [12,13].

Knowledge about the role of TrxR in GBM pathogenesis is still limited. A single study has documented an elevated TrxR activity in GBM compared to normal brain tissue [14]. Recently, an association between a high level of cytoplasmic TrxR and adverse overall survival of GBM patients has been shown [15]. Those findings suggest that TrxR may be an attractive drug target in GBM. This, in turn, draws attention to AF as a clinically available inhibitor of the Trx system that could be used in the treatment of this cancer. Noteworthy, in a phase Ib/IIa trial (NCT02770378) completed with promising results, AF was one of nine drugs in the treatment protocol which used re-purposed, older drugs as an add-on treatment to the standard-of-care temozolomide, for recurrent GBM [16]. In recent studies, increasing of oxidative stress level either via a biochemical activation of small-conductance K_Ca_ (SK) channels, or an application of cold atmospheric plasma, potentiated AF-induced GBM cell death [17,18].

In this study, we analyzed the expression of the cytosolic isoform of TrxR (TrxR1), both at the protein and mRNA level, in human GBM and non-tumor (NT) brain tissues. Next, the influence of AF on viability and clonogenic potential of human GBM cell lines and patient-derived GBM cells was examined. Moreover, we analyzed the effect of AF on the ability of patient-derived GBM cells to generate neurospheres which is considered a surrogate assay for GSC self-renewal. In addition, we studied the cellular response upon combination treatment of AF with menadione (2-methyl-1,4-naphthoquinone; MEN, vit-amin K3), whose anticancer activity is related to ROS production [19]. Furthermore, we examined the influence of treatment with AF alone or in combination with MEN on viability of normal human astrocytes (NHA).

## 2. Results

### 2.1. Expression of TrxR1 in NT and GBM Tissues and Cells

To broaden the knowledge about the expression of TrxR1 in brain tumors, we ana-lyzed the publicly available datasets for the gene expression of *TXNRD1* mRNA in GBM and NT brain tissues. As shown in Figure 1a, in each of three datasets examined, transcripts for *TXNRD1* were markedly elevated in GBM when compared to the NT specimens. Furthermore, we examined the expression of *TXNRD1* at mRNA and protein level in 12 NT and 24 GBM tissues. A tendency toward increased *TXNRD1* expression was detected in GBM compared to NT specimens (Figure 1b). At the protein level, subtle differences in the amount of TrxR1 were observed between particular NT tissues, and substantial variability was found among GBM specimens (Figure 1c, left panel). Densitometric analysis revealed the trend of increasing TrxR1 amount in GBM compared to NT tissues (Figure 1c, right panel). Overall, considerable expression of TrxR1 in most of GBM tissues both at the mRNA and protein level indicates that the Trx system could be an attractive drug target.

In further studies we used T98G, U87MG, and LN229 GBM cell lines. Since commercially available cell lines do not fully reflect GBM heterogeneity which significantly contributes to therapeutic failure, we also used LUB17 and LUB20 cell cultures derived from GBM tissues #11 and #12, respectively, grown as adherent monolayers. Additionally, cells isolated from GBM #11 were cultured as neurospheres (hereinafter referred to as LUB17N). Substantial amounts of TrxR1 protein were found in each of the GBM cell lines, while only traces of this protein were detected in NHA (Figure 1d).

### 2.2. The Influence of AF on Viability of GBM Cells and NHA

To investigate the influence of AF on the viability of GBM cells and NHA, adherent cell cultures were incubated with increasing concentrations of AF for 72 h. Treatment with 0.25 μM AF reduced viability of commercially available cells by 40–55%. The patient-derived GBM cells turned out to be more susceptible to 0.25 μM AF, since such a treatment lowered their viability by approximately 70%. The highest concentration of AF used in this study (1 μM) decreased viability of U87MG cells to 25%, and to 10–15% in case of the other cell lines. NHA were less vulnerable than GBM cells to AF treatment, since the decrease in their viability did not exceed 50% even upon treatment with 1 μM AF. It should be mentioned that all GBM cells used in this study were significantly more sensitive than NHA to treatment with both 0.5 μM and 1 μM concentrations of AF. Furthermore, LN229, LUB17, and LUB20 cells were also found to be more vulnerable than NHA to the effects of 0.25 μM AF (Figure 2).

### 2.3. The Influence of AF on Ability of GBM Cells to Form Colonies and Neurospheres

Next, we investigated the influence of long-term treatment with AF on the ability of GBM cells to form colonies. The clonogenic potential of the U87MG and LN229 cells with 0.25 μM for 72 h was significantly lower compared to the DMSO-treated cells, and the reduction of this parameter was also observed in T98G cells. Treatment with higher concentrations of AF further decreased the number of colonies formed by T98G, U87MG, and LN229 cells. Of note, similar to the results obtained in the MTT assay, the clonogenic assay patient-derived cells turned out to be more sensitive to treatment with AF than the commercially available cells (Figure 3a). Furthermore, the ability of LUB17 cells to generate neurospheres, a surrogate marker of GSC self-renewal, was also significantly affected by the presence of 0.25 and 0.5 μM AF and almost completely inhibited by treatment with 1 μM AF for 20 days (Figure 3b). Combined, these data indicate cytotoxic effect of treatment with AF on GBM cells cultured both as adherent monolayers and neurospheres.

### 2.4. AF-Induced ROS Production Contributes to GBM Cell Death

Next, we studied the mechanism of cell death triggered by AF. Since this compound has been reported to increase intracellular ROS level [18,20], we hypothesized that AF-induced ROS production contributed to GBM cell death. Indeed, in all cell lines, treatment with 1 μM AF for 24 h significantly increased intracellular ROS level estimated by using DCFDA as a probe (Figure 4a). Pre-treatment with the ROS scavenger NAC prevented the AF-induced cell death (Figure 4b), supporting the driving influence of ROS in the reduction of cell viability.

### 2.5. Effect of Treatment with AF on the Susceptibility of GBM Cells and NHA to MEN

Given the increased ROS levels observed in GBM cells treated with AF, we hypothe-sized that combined treatment with AF and a ROS inducer MEN would potentiate cell death. Indeed, treatment with 10 μM MEN for 72 h reduced the viability of U87MG and LN229 cells by approximately 20%, while it reduced the viability of LUB20 cells by 25%. T98G and LUB17 cells appeared to be more sensitive to 10 μM MEN, since after 72 h their cell viability achieved 62% and 58%, respectively (Figure 5a). Combined treatment with 10 μM MEN and either 0.25 μM or 0.5 μM AF for 72 h resulted in a strong reduction of the cell viability. Treatment with 0.25 μM AF and 10 μM MEN decreased this parameter to approximately 10% in all cell lines except U87MG, which showed more than 30% of viable cells. Treatment with 0.5 μM AF and 10 μM MEN enhanced cell death, although U87MG cells remained less susceptible than the other cell lines used in this study (Figure 5a). Moreover, combined treatment with 10 μM MEN and either 0.25 μM or 0.5 μM AF for 72 h exhibited the least cytotoxicity towards NHA, since in these conditions’ viability of NHA achieved 60% and 48%, respectively (Figure 5a).

Next, we examined the influence of the combined treatment with AF and on the gen-eration of neurospheres by LUB17N cells. Application of 10 μM MEN alone for 20 days reduced the ability to form neurospheres by approximately 30%. Upon treatment with 0.25 μM and 10 μM, MEN neurosphere generation was reduced to 15% and almost completely abolished when a combination of 0.5 μM AF and 10 μM MEN was used (Figure 5b).

Collectively, these results indicate that combination of AF and MEN enhances the cytotoxic activity toward GBM cells presented by each of these compounds applied alone.

## 3. Discussion

Oxidative stress perturbs cell health and leads to cell death. Thus, oxidative stress-inducing agents have attracted much attention as potential therapeutic anticancer strategies. However, cancer cells generate high levels of ROS as by-products of their metabolism. Therefore, to survive, they increase their antioxidative capacity which facilitates adaptation to excessive ROS conditions. This, in turn, may result in the resistance to the redox-targeted approaches [21]. The combination of ROS inducer either with another ROS inducer, or with antioxidant silencer has been proposed as the strategy to overcome the ROS adaptation mechanisms [22]. The identification of the crucial adaptation mechanisms which neoplastic cells have developed to combat excess ROS becomes an essential step in designing anticancer therapy based on oxidative damage.

In this study we focused on TrxR1, regarded as one of the master regulators of cellular redox homeostasis, due to its central role in cytosolic H_2_O_2_ detoxification. Elevated expression and/or activity of this protein has been observed in several cancers [23]. In the current study, high expression of TrxR1 both at the mRNA and protein level was detected in the majority of GBM patient group. This finding remains in accordance with observations by Kemerdere et al. who reported elevated TrxR activity in GBM tissues compared to the control group [14]. Moreover, a recent study suggested an association between high expression of cytoplasmic TrxR and poor overall survival of GBM patients [15]. Collectively, these observations point to the pathophysiological role of TrxR in GBM and the therapeutic potential of its inhibitor, AF.

Keeping the above in mind, we decided to evaluate the activity of AF in three commercially available human GBM cell lines: T98G, U87MG, and LN229. Among them, T98G cells were the most resistant to the lowest, 0.25 μM concentration of AF used in this study. When AF was applied in the higher concentration of 1 μM, U87MG cells turned out to be more resistant. These observations are partially consistent with the previous study, in which resistance to AF presented by T98G cells correlated with higher GSH baseline levels and TrxR activity [18]. Indeed, here we have shown that out of three cell lines used, T98G cells contain the highest amount of TrxR1 protein. Worth mentioning, two patient-derived GBM primary cultures were more susceptible to AF than any of the cell lines, which could also result from the lower amount of TrxR1 protein. Moreover, those cells appeared to be most sensitive to AF treatment in a clonogenic assay. Interestingly, patient-derived cells referred to as LUB17 grown in neurospheres were more resistant to AF than their counterparts cultured as an adherent monolayer. This finding is in line with the study by Van Loenhout et al. showing that adherent T98G, U87MG, and LN229 cells are more sensitive to AF compared to the relevant cells cultured in neurospheres [18]. Although the reasons of this phenomenon remain unknown, the speculation was made that GBM spheroids adapted to the increased intracellular ROS levels by enhancing their protective antioxidant system. Indeed, Khaitan et al. showed that GBM spheroids were more resistant to radiation and presented lower radiation-induced oxidative stress than GBM cells cultured in monolayer [24]. Our results seem to support this hypothesis, since LUB17N neurospheres contain a higher amount of TrxR1 protein compared to LUB17 adherent cells.

AF treatment increased the intracellular ROS levels in all three cell lines and both patient-derived cell cultures examined. Worth mentioning, ROS overproduction was most pronounced in LUB20 cells, presenting the highest susceptibility to AF short-term treatment. Pre-treatment with a ROS scavenger NAC partially prevented AF-induced cell death, confirming the driving influence of the excess of ROS in the reduction of cell viability. These results corroborate the findings of earlier studies conducted on the different models in which antineoplastic activity of AF has been shown to be closely related with ROS accumulation [18,20,25].

Finally, we decided to examine the effects of AF applied together with MEN on GBM cell viability. Cytotoxic activity of MEN, related to ROS production during its metabolism, has been documented in cancer cell lines of different origin, including GBM cells in which it was also examined in combination with ascorbic acid (AA) [19,26,27]. Nevertheless, to the best of our knowledge, there are no studies on anticancer activity of AF combined with MEN. Here, we show that 10 μM MEN alone slightly decreases viability of both GBM cell lines and patient-derived cells. However, this parameter is dramatically reduced upon treatment composed of both MEN and AF. Moreover, combination of these two drugs displays highly potent efficacy against GBM neurospheres generation. It must be emphasized, that according to literature, both AF and MEN should be able to cross the blood–brain barrier [26,28], one of the major obstacles for treatment of brain tumors. Additionally, numerous studies on these two compounds indicate their safety. Indeed, in our study, NHA displayed lower sensitivity to treatment with AF compared to GBM cells. This finding further corroborates relatively low cytotoxicity of AF towards non-tumor cells. It is worth mentioning that in addition to AF, a number of gold-containing TrxR1 inhibitors have been identified [29,30]. Although the detailed mechanism of inhibition of TrxR1 activity displayed by those compounds is still unclear, they undoubtedly deserve further investigation.

Together, our results obtained on several GBM cell lines and patient-derived GBM cells warrant further investigations on AF in combination with MEN to exploit their potential application in the treatment of GBM.

## 4. Materials and Methods

### 4.1. Clinical Material

Twenty-four GBM tissues were collected from the Department of Neurosurgery and Paediatric Neurosurgery, at the Medical University of Lublin, Poland. Prior written informed consent was obtained from all patients in accordance with the Declaration of Helsinki, and the local Ethics Committee approved all procedures. Histological examination was performed according to the WHO criteria [31]. Twelve NT brain samples were obtained from The Netherlands Brain Bank (NBB), Netherlands Institute for Neuroscience, Amsterdam (open access: www.brainbank.nl). All material has been collected from donors for or from whom a written informed consent for a brain autopsy and the use of the material and clinical information for research purposes had been obtained by the NBB.

### 4.2. GBM Cell Lines and Human Astrocytes Culture

T98G human GBM cell line (American Type Culture Collection, Manassas, VA, USA) was maintained in Earle’s Minimal Essential Medium (MEME) (Sigma-Aldrich, St. Louis, MO, USA) supplemented with 10% fetal bovine serum (FBS) (Gibco, Thermo Fisher Scientific, Grand Island, NY, USA), non-essential amino acids (Gibco) and 1% antibiotics (penicillin and streptomycin) (Gibco).

U87MG human GBM cell line (Sigma-Aldrich) was maintained in Eagle’s Minimum Essential Medium (EMEM) (ATCC) supplemented with 15% FBS and 1% antibiotics (penicillin and streptomycin) (Gibco).

LN229 human GBM cell line (a kind gift from Rafał Krętowski, PhD, Department of Pharmaceutical Biochemistry, Medical University of Białystok, Białystok, Poland) was maintained in Dulbecco’s Modified Eagle Medium (DMEM) (Gibco) supplemented with glucose (final concentration 4.5 g/L), 10% FBS and 1% antibiotics (penicillin and streptomycin) (Gibco).

All cell lines were maintained at 37 °C in a humidified atmosphere with 95% air and 5% CO_2_ and tested regularly for mycoplasma contamination using Mycoplasma Detection Kit-Quick Test (Biotool, Stratech Scientific Ltd., Cambridge, UK).

Normal human astrocytes (NHA) were purchased from ScienCell Research Laboratories (Carlsbad, CA, USA) and cultured in the Astrocyte Medium according to the manufacturer’s instruction 37 °C, 5% CO_2_ and 95% humidity.

### 4.3. Primary GBM Cell Cultures

Patient-derived GBM cell lines were established according to the protocol by Mullins et al. [32]. Briefly, fresh tumor tissues were minced by scalpel in DMEM/HAM’s F12 medium (Gibco), supplemented with 10% FBS and 1% penicillin-streptomycin (Gibco) and passed through a cell strainer to obtain a single-cell suspension and seeded in plates. Outgrowing cells were detached with TrypLE Express Enzyme (Thermo Fisher Scientific) and expanded for subsequent analyses.

Patient-derived GBM neurospheres were established according to the protocol by Wu et al. [33]. Briefly, cells were cultured in DMEM/HAM’s F12 medium supplemented with growth factors, including 10 ng/mL basic Fibroblast Growth Factor (bFGF), 20 ng/mL Epidermal Growth Factor (EGF), and B-27 (1:50 dilution). All growth factors were purchased from Gibco.

### 4.4. Analysis of Publicly Available Datasets

Microarray data from the following publicly available databases were utilized: REpository for Molecular BRAin NeoplasiaDaTa (REMBRANDT) [34] (NT = 28; GBM = 219), The Cancer Genome Atlas (TCGA) (https://www.cancer.gov/tcga (accessed on 9 August 2022)) (NT = 10; GBM = 528), and GSE16011 dataset [35] (NT = 8; GBM = 159). The datasets were accessed from the GlioVis web application (http://gliovis.bioinfo.cnio.es (accessed on 9 August 2022)) [36].

### 4.5. RNA Isolation, RT-PCR, and Real-Time PCR

Total RNA from the tissues or cells was extracted using TRI-Reagent (Sigma-Aldrich), according to the manufacturer’s protocol. RNA concentration was measured using NanoDrop2000 and two µg of RNA were reverse-transcribed using a High Capacity cDNA Reverse Transcription Kit (Applied Biosystems, Thermo Fisher Scientific) according to the manufacturer’s protocol. The *TXNRD1* primers were purchased from Applied Biosystems (assay ID: Hs00917067_m1), and β-actin primers were purchased from Blirt (Gdansk, PL) (cat no HK-DD-hu). Each reaction (total volume of reaction 10 µL) contained 5 µL TaqMan Universal PCR Master Mix (Applied Biosystems, Thermo Fisher Scientific), 1 µL cDNA, 0.5 µL primers and RNAse/DNAse free water. The real-time PCR reactions were performed at 95 °C for 10 min, followed by 40 cycles of 30 s at 95 °C and 1 min at 60 °C. Relative expression was calculated using the ΔΔCT method [37] and normalized to the expression of β-actin.

### 4.6. Protein Isolation and Western Blot

Total protein was extracted using ice-cold RIPA buffer (Sigma-Aldrich) with protease and phosphatase inhibitors (Sigma-Aldrich). Protein concentrations were determined by bicinchoninic acid Protein Assay Kit (Pierce, Rockford, IL, USA). Ten µg of proteins were separated by 10% SDS-PAGE and then electrotransferred to polyvinylidene difluoride membranes. Membranes were blocked with 5% skim milk for 1 h at room temperature, and incubated with a human-specific anti-TrxR1 antibody (ProteinTech, Chicago, IL, USA) overnight at 4 °C. After washing, membranes were incubated with anti-rabbit antibody (Sigma-Aldrich) conjugated to horseradish peroxidase for 1 h at room temperature. Blots were visualized on X-ray film using a SuperSignal West Pico Chemiluminiscence Substrate (Pierce). For loading control, membranes were stripped and reused with an antibody against β-actin (ProteinTech). Densitometric analysis was performed using ImageJ software (National Institutes of Health, Bethesda, MD, USA) to calculate TrxR1/β-actin ratio.

### 4.7. Chemicals

AF was purchased from Santa Cruz Biotechnology (Dallas, TX, USA). MEN and antioxidant N-acetylcysteine (NAC) were purchased from Sigma-Aldrich. All compounds were dissolved in dimethyl sulfoxide (DMSO) (Sigma-Aldrich), according to the manufacturers’ instructions. In all assays, control groups were treated with DMSO. The final concentration of DMSO did not exceed 0.1% *v/v* and did not affect cell growth or death.

### 4.8. Cell Viability Assessment

Cell survival was determined by 3-(4,5-dimethylthiazol-2-yl)-2,5-diphenyl tetrazolium bromide (MTT) conversion into formazan. The cells were seeded into 24-well plates (4 × 10^4^ per well) and incubated overnight at 37 °C. After this time, the cells were treated with increasing concentrations (0, 0.25, 0.5, 1 μM) of AF, 10 μM MEN, and their combination for 72 h. In some experiments, the cells were pre-incubated with 5 mM NAC for 1 h prior to exposure to AF. Subsequently, the cells were washed with phosphate-buffered saline (PBS) (Sigma-Aldrich) and incubated in the culture medium with MTT (Sigma-Aldrich) solution at the final concentration of 0.5 mg/mL for 2 h at 37 °C. Then, the medium was replaced with 100% DMSO and absorbance was read at 570 nm using Elisa Bio-Rad Microplate Reader (Bio-Rad, Hercules, CA, USA). Each compound in each concentration was tested in triplicate in a single experiment and at least three independent experiments were performed.

### 4.9. Clonogenic Assay

T98G, U87MG, LN229, U87MG, LUB17, or LUB20 cells were seeded in six-well plates (T98G, U87MG, LN229, and LUB19 cells: 200 per well; LUB20 cells: 500 per well) and allowed to attach for 24 h. Next, cells were treated with increasing concentrations (0.25, 0.5, and 1 μM) of AF or DMSO (Sigma-Aldrich) for 72 h. After this time, the medium was replaced with drug-free medium, and cells were incubated for 10 days, during which time the medium was changed regularly. Next, cell culture plates containing colonies were gently washed with PBS (Sigma-Aldrich) and fixed with 4% formaldehyde for 10 min. Colonies were stained with 0.5% crystal violet solution in 25% methanol for 10 min and the grossly visible colonies were counted manually. Three independent experiments were performed.

### 4.10. Neurosphere Generation

LUB17N cells were seeded in 24-well plates. After 24 h, cells were treated with increasing concentrations (0.25, 0.5, 1 μM) of AF, 10 μM MEN, and their combination for 20 days and neurosphere size was evaluated. Briefly, 5 bright field images at 4× magnification were randomly taken from each condition with an inverted phase contrast microscope Axiovert 40C (Zeiss, Oberkochen, Germany) and analyzed using ImageJ software. The neurosphere average area, as expressed in mm^2^, was calculated dividing the entire neurosphere area by the total number of neurospheres.

### 4.11. Evaluation of ROS Generation

ROS measurement was performed using the cell-permeable fluorescent probe 2′,7′- dichlorodihydrofluorescein diacetate (H_2_DCFDA) (Sigma-Aldrich). Briefly, the cells were seeded in 96-well plates and allowed to adhere overnight. Next, they were exposed to various concentrations of AF for 24 h and then loaded with H_2_DCFDA (10 μM). Following incubation at 37 °C for 30 min, the cells were washed with PBS and fluorescence intensity was measured (excitation = 485 nm; emission = 530 nm) using a microplate reader (Tecan, Infinite M2000). Data is presented as fold change from DMSO-treated cells.

### 4.12. Statistical Analysis

All data were analysed using Prism 7.0 GraphPad Software (San Diego, CA, USA). Results are expressed as mean ± SEM from at least three independent experiments. For comparison between two means, the Mann–Whitney test was used. For comparisons of the mean values among groups, a one-way ANOVA was used.

## 5. Conclusions

In conclusion, the present study demonstrates considerable amounts of TrxR1 isoform in GBM tissues and therefore points to the therapeutic potential of its inhibitor, AF. Furthermore, our work documents the high effectiveness of AF in eradicating several GBM cell lines as well as patient-derived GBM cells grown both as monolayer or as neurospheres. Finally, it also shows that MEN potentiates AF-induced reduction of GBM cell viability. Altogether, our study supports further preclinical evaluation of AF in combination with MEN as a novel approach for GBM treatment.

## Figures and Tables

**Figure 1 ijms-23-15712-f001:**
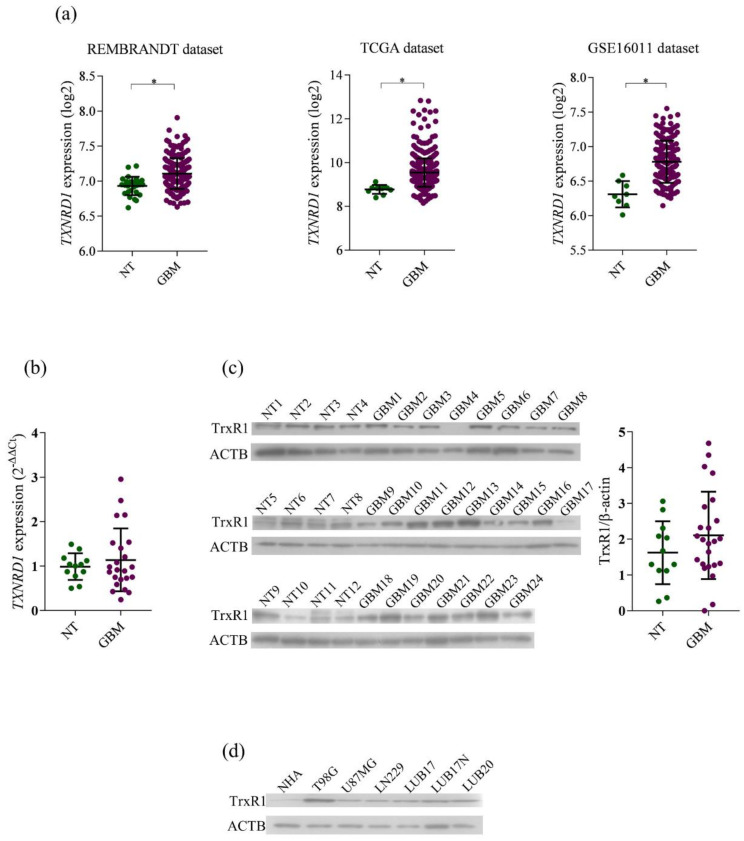
Characterization of TrxR1 expression level in human GBM and non-tumor brain tissues, human astrocytes and GBM cells. (**a**) Analysis of the expression of *TXNRD1* mRNA in non-tumorigenic (NT) and GBM tissues available in REMBRANDT datasets (NT = 28; GBM = 219), TCGA (NT = 10; GBM = 528), and GSE16011 (NT = 8; GBM = 159). Statistical analysis was performed with Mann–Whitney test (* *p* < 0.0001). (**b**) Analysis of the expression of *TXNRD1* mRNA in human NT (n = 12) and GBM (n = 24) tissues. (**c**) Western blot (left panel) and densitometric analysis (right panel) of TrxR1 protein in human NT (n = 12) and GBM (n = 24) tissues. (**d**) Analysis of the level of TrxR1 protein in normal human astrocytes (NHA), human GBM cell lines (T98G, U87MG, and LN229) and human GBM cell cultures grown as adherent monolayers (LUB17, LUB20) or as neurospheres (LUB17N).

**Figure 2 ijms-23-15712-f002:**
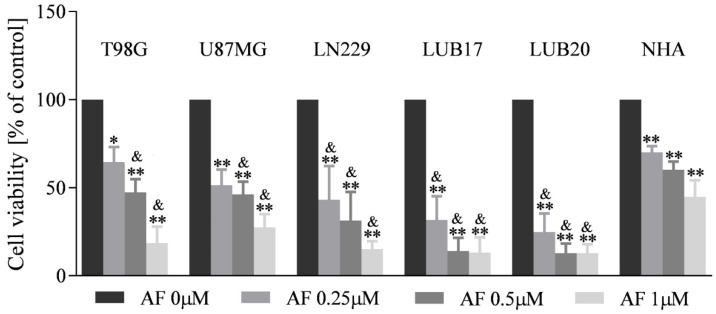
The influence of AF on the viability of GBM cells and NHA. Cells were treated with increasing concentrations (0, 0.25, 0.5, 1 μM) of AF for 72 h. Subsequently, MTT assay was performed. Viability of the cells is shown as percentage of the control cells treated with DMSO. An experiment was performed in triplicates and repeated at least three times. Statistical analysis was performed with one-way ANOVA. * *p* < 0.05, ** *p* < 0.005 vs. DMSO-treated cells; and & *p* < 0.05 vs. NHA treated the same way.

**Figure 3 ijms-23-15712-f003:**
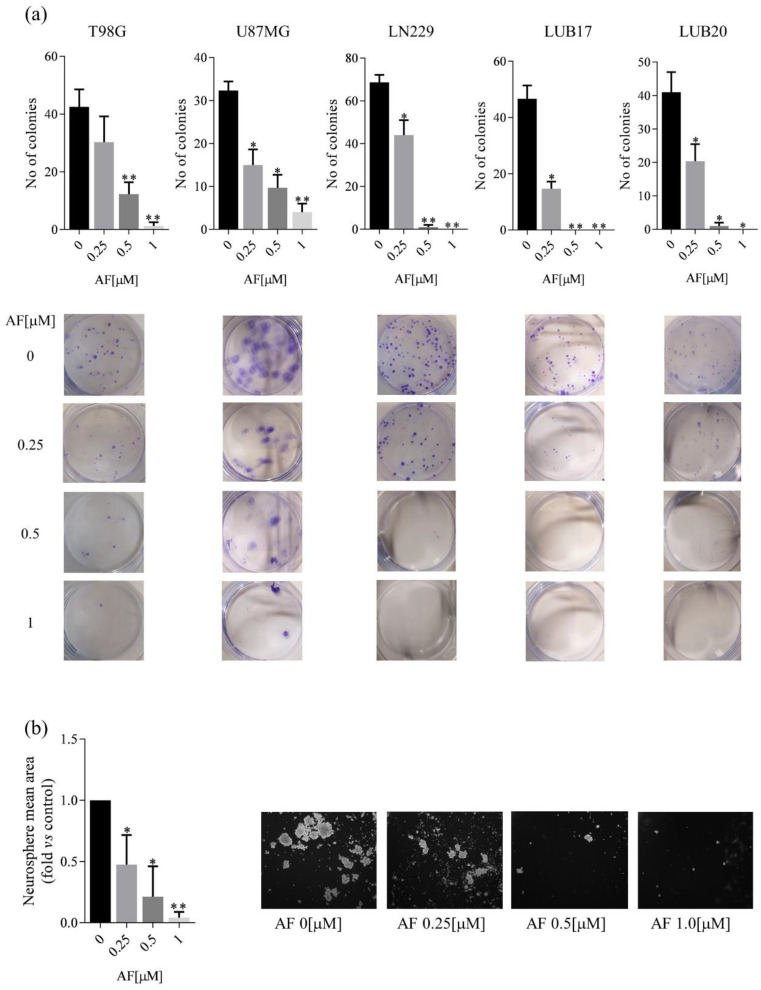
Treatment with AF inhibits the clonogenic potential of GBM cells and their ability to generate neurospheres. (**a**) The results of clonogenic assay expressed as total number of colonies per well formed after treatment for 72 h with increasing concentrations (0, 0.25, 0.5, 1 μM) of AF and subsequent incubation in drug-free medium for 10 days (upper panel) and representative images (lower panel). (**b**) Quantification analysis of neurosphere mean area expressed as fold vs. DMSO-treated cells (left panel) and representative phase-contrast images (4 x magnification) (right panel) of LUB17N cells maintained in the absence or presence of increasing concentrations of AF for 20 days. Experiments were repeated at three times. Statistical analysis was performed with one-way ANOVA. * *p* < 0.05, ** *p* < 0.005.

**Figure 4 ijms-23-15712-f004:**
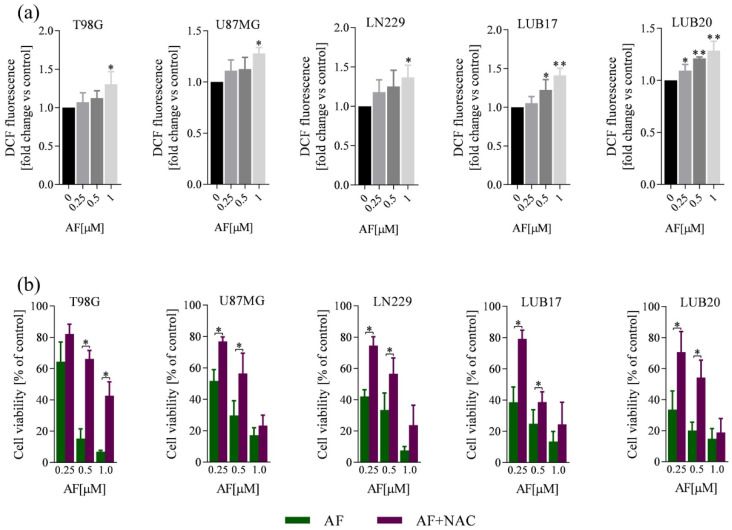
AF-induced increase of intracellular ROS level contributes to the reduction of the viability of GBM cells. (**a**) Intracellular DCF fluorescence intensity of cells treated with increasing concentrations (0, 0.25, 0.5, 1 μM) of AF for 24 h. (**b**) Viability of the cells treated with increasing concentrations (0, 0.25, 0.5, 1 μM) of AF for 72 h with or without pre-treatment with 5 mM NAC for 1 h. Experiments were performed in triplicates and repeated three times. Statistical analysis was performed with one-way ANOVA. * *p* < 0.05, ** *p* < 0.005.

**Figure 5 ijms-23-15712-f005:**
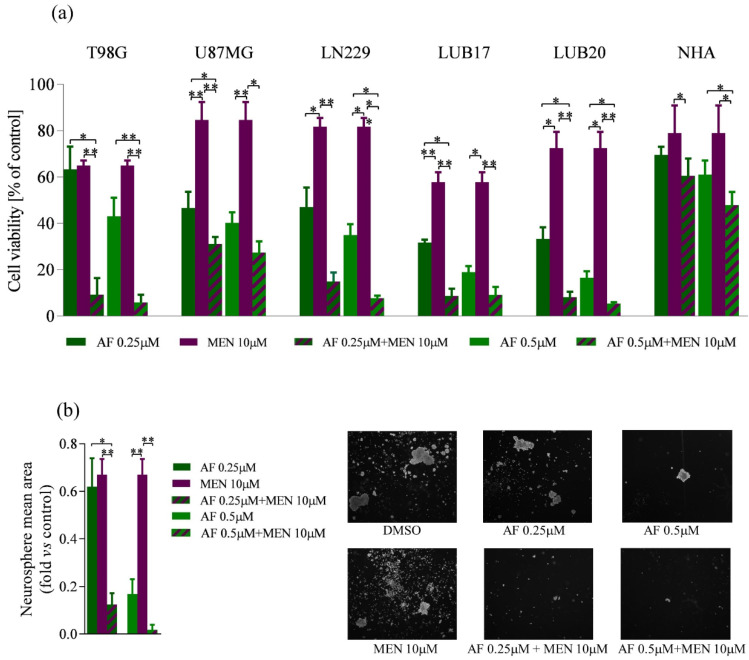
The influence of the combined treatment of AF and MEN on GBM cells and NHA viability and neurosphere formation. (**a**) Adherent cells were treated with either 0.25 μM, 0.5 μM AF, or 10 μM MEN alone or in combination for 72 h. Subsequently, MTT assay was performed. Viability of the cells is shown as percentage of the control cells treated with DMSO. (**b**) Quantification analysis of neurosphere mean area expressed as fold vs. DMSO-treated cells (left panel) and representative phase-contrast images (4× magnification) (right panel) of LUB17N cells maintained in the absence or presence of 0.25 μM, 0.5 μM AF, or 10 μM MEN alone or in combination for 20 days. Experiments were performed in triplicates and repeated at least three times. Statistical analysis was performed with one-way ANOVA. * *p* < 0.05, ** *p* < 0.005.

## Data Availability

Publicly available datasets were analyzed in this study. This data can be found here: TCGA dataset (generated by the TCGA Research Network, https://www.cancer.gov/tcga (accessed on 9 August 2022)); https://portal.gdc.cancer.gov (accessed on 9 August 2022); REMBRANDT dataset: https://www.ncbi.nlm.nih.gov/geo/query/acc.cgi?acc=GSE108476 (accessed on 9 August 2022); GSE16011 dataset: https://www.ncbi.nlm.nih.gov/geo/query/acc.cgi?acc=gse16011 (accessed on 9 August 2022).

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
