# Peer review of "Menadione Potentiates Auranofin-Induced Glioblastoma Cell Death"

_ijms, 2022, doi:10.3390/ijms232415712_

Round 1
Reviewer 1 Report
In this manuscript, Szeliga and Rola determined the influence of an inhibitor of TrxR1, auranofin (AF), alone or in combination with a prooxidant menadione (MEN), on growth of GBM. These results are promising and indicate that AF used alone or in combination with MEN may serve as an alternative therapeutic regimen. However, additional experiments and clarification are recommended to provided before publication.
1. In Figure 1b, the western blotting signal of ACTB in line 2 seems different, which means the whole protein loading amount is different. Therefore, I would suggest remeasuring it to make it more convincing.
2. Since blood brain barrier is the major obstacle that prevents drug penetration, therefore, the efficiency of AF and MEN to cross BBB is recommended to evaluated.
3. According to the MTT assay, AF and MEN are able to induce high cytotoxicity to either commercial cell line or patient-derived GBM. However, the cytotoxicity of AF and MEN on normal cell should also be tested, in order to evaluate the safety.
4. Reference (Chin. Chem. Lett., 2022, 33(7), 3488-3491) is recommended to be cited.
Reviewer 2 Report
The manuscript (ijms-2014451) provided by Szeliga et al. discusses the use of a combination of auranofin, a TrxR1 inhibitor, with pro-oxidative menadione. In addition, the authors analyzed the expression of TrxR1 mRNA and protein in GBM tissues. The work has great potential in the context of the use of drugs influencing oxidative stress in cancer cells, as well as drug repositioning. The studies showed high cytotoxicity of AF, as well as the combination of AF and MEN on GBM cells. This confirms that TrxR1 may be a target in the treatment of glioblastoma. This research is very detailed and innovative, the results of this work might be of interest to the readers of IJMS. However, the reviewer has a few reservations and comments that require clarification and correction.
1. Have AF, MEN, and their combinations been studied in the context of cytotoxicity to normal human astrocytes (NHA)? At least the results of the AF and MEN cytotoxicity test should be included in the manuscript.
2. The reviewer has great reservations about the results of the Western Blots shown in Figures 1b and 1c. In the presented Western Blots, there are substantial differences in the intensity of the bands for the reference protein- beta-actin. This means that the protein concentration in the used samples was different. It is incorrect to conclude about the amount of TrxR1 when samples with different concentrations are applied.
3. Clonogenic assay also needs clarification. Were the GBM cells treated with the tested compound throughout the incubation time which means 14 days? Was the culture medium changed during these 14 days? How were the cell colonies counted in the U97MG line? The photo from the well shows that the colonies are very large and fused together. What was the colony counting method? Was any computer program used or was it done manually?
4. Why was the MTT test performed after 72 hours of incubation with the compound, but the ROS generation assay after 24 hours?
5. Subsections 2.1, 2.2, and 2.4 have the same headings.
6. What was the number of cells seeded per well for the clonogenic assay and MTT?
Best regards
Round 2
Reviewer 2 Report
Response to Reviewer #2 comments:
We thank Reviewer #2 for the careful and insightful review of our manuscript. We have addressed all of the concerns as summarized below:
1. Have AF, MEN, and their combinations been studied in the context of cytotoxicity to normal human astrocytes (NHA)? At least the results of the AF and MEN cytotoxicity test should be included in the manuscript.
Response: We thank the Reviewer for this suggestion. We examined the influence of AF alone and in combination with MEN on the viability of normal human astrocytes. The results are described in lines 114-115 and 171-174 and presented in Figure 2 and 5, respectively.
Reviewer response: The addition of studies on AF and MEN toxicity on NHA cells significantly supplemented the manuscript. However, there is no discussion of these results in the discussion section. Maybe there are some conclusions about which cell line is the best target for AF as compared to the results obtained on NHA?
2. The reviewer has great reservations about the results of the Western Blots shown in Figures 1b and 1c. In the presented Western Blots, there are substantial differences in the intensity of the bands for the reference protein- beta-actin. This means that the protein concentration in the used samples was different. It is incorrect to conclude about the amount of TrxR1 when samples with different concentrations are applied.
Response: We thank the Reviewer for pointing out this error. We re-measured the amount of total protein in the samples and then performed additional western blot analysis. The results are presented in Figure 1b and 1c.
Reviewer response: Unfortunately, the reviewer still has reservations about the thickness of the bands corresponding to beta-actin as the loading control. Only in Fig. 1b set 2 the bands corresponding to beta actin are comparable in each sample. In Figure 1b set 1 there is a clear difference between the band in GBM #4 and GBM #5 samples for beta-actin. This may lead to the consideration that the amount of TrxR1 in GBM #4 could be greater if the beta-actin band looked like in GBM #5. The reviewer suggests densitometric reading of beta actin and TrxR1 bands and making ratios that will indicate the amount of TrxR1 in all tested samples. Similar differences exist in Figure 1b set 3 for the GBM #23 and GBM #24 samples.
3. Clonogenic assay also needs clarification. Were the GBM cells treated with the tested compound throughout the incubation time which means 14 days? Was the culture medium changed during these 14 days? How were the cell colonies counted in the U97MG line? The photo from the well shows that the colonies are very large and fused together. What was the colony counting method? Was any computer program used or was it done manually?
Response: Thank you for bringing this issue to our attention. The cells were treated with required concentrations of AF or DMSO throughout the whole incubation time of 14 days. The medium was regularly changed during the incubation time and AF was added with each change of the medium. The colonies were counted manually. To clarify this issue, an appropriate description has been added to the methodological section 4.9 (Clonogenic assay). We fully agree that U87MG cells form colonies which are often fused together and therefore the counting process of those particular colonies is challenging. We did it manually, as in the case of the other cell lines. However, we also applied another often used technique: colonies stained with crystal violet were dissolved in 1% SDS, and the optical density was measured at 570 nm using a microplate reader. Since there are some steps in this procedure during which additional errors may be generated (e.g. inaccurate dissolving, pipetting error during transferring to microtitration plate), we do not use it routinely. Nevertheless, we applied this procedure to verify the results obtained during counting U87MG colonies. Below we attach the results of this analysis (Fig. 1). Please note that in general the differences in OD values for the particular treatments reflect the differences observed in clonogenic assay.
Reviewer response: The author's answer indicates that with each change of medium during clonogenic assay, AF was added anew, which means that for 14 days, AF was added to the cells at least 4 times. Unfortunately, according to the reviewer's knowledge, such methodology of this experiment is incorrect and unacceptable.
According to the knowledge of the reviewer and most scientific articles in highly scored journals, the clonogenic assay methodology is consistent with the method described, for example, in the following articles: Petsri et al. 2022 DOI: 10.1186/s12906-022-03727-6; Franken et al. 2006 DOI: 10.1038/nprot.2006.339; Ascer et al. 2015 DOI: 10.1002/jcb.25166 and many more.
In a properly performed clonogenic assay method, the cytotoxic agent should be added to tested cells for the standard incubation time (in the case of the author`s research it would be 72h), and then the medium containing AF should be removed and replaced with the fresh, clean medium. Cell medium should be replaced every two days with the fresh, clean medium without compound throughout the all incubation time (14 days).
Adding AF several times during the clonogenic assay could cause the accumulation of this compound in the cells and, in fact, a much higher concentration of this compound inside the tested cells.
The reviewer advises performing this experiment again using the correct methodology.
As for the colony counting method, the reviewer accepts the authors' explanations.
4. Why was the MTT test performed after 72 hours of incubation with the compound, but the ROS generation assay after 24 hours?
Response: We assumed that treatment with AF would result in an increase in ROS levels, which in turn would lead to cell death. Given this cause-effect relationship, an increase in ROS levels should occur before cell death. Therefore, we examined the ROS levels after 24h treatment with AF (when most of the cell were still alive), while cell death after 72h of treatment with AF. Please note that the similar procedure was used in several other studies assuming the contribution of an elevated ROS production to cell death, including the papers cited in our manuscript: Van Loenhout et al., Cells. 2021, 10, 2936. doi: 10.3390/cells10112936; Karsa et al., Br J Cancer. 2021, 125, 55-64. doi: 10.1038/s41416-021-01332-x.
Reviewer response: The reviewer has no additional comments and accepts the authors' explanations.
5. Subsections 2.1, 2.2, and 2.4 have the same headings.
Response: The headings of subsections have been modified.
Reviewer response: The reviewer has no additional comments.
6. What was the number of cells seeded per well for the clonogenic assay and MTT?
Response: For the clonogenic assay T98G, U87MG, LN229 and LUB17 cells were seeded at 200 cells/well, while LUB20 were seeded at 500 cells/well. For the MTT assay, 4x10 4 cells/well were seeded. An appropriate description has been added to the methodological sections 4.8 (Cell viability assessment) and 4.9 (Clonogenic assay).
Reviewer response: The reviewer has no additional comments.
